# A Sliding Window-Based CNN-BiGRU Approach for Human Skeletal Pose Estimation Using mmWave Radar

**DOI:** 10.3390/s25041070

**Published:** 2025-02-11

**Authors:** Yuquan Luo, Yuqiang He, Yaxin Li, Huaiqiang Liu, Jun Wang, Fei Gao

**Affiliations:** 1School of Electronic Information Engineering, Beihang University, Beijing 100191, China; luoyuquanhz@163.com (Y.L.); buaahyq@buaa.edu.cn (Y.H.); liuhuaiqiang@lyu.edu.cn (H.L.); wangj203@buaa.edu.cn (J.W.); feigao2000@163.com (F.G.); 2Hangzhou Innovation Institute of Beihang University, Hangzhou 310052, China

**Keywords:** mmWave radar, point cloud, skeletal pose estimation, multi-frame time-series data, convolutional neural network, bidirectional gated recurrent unit

## Abstract

In this paper, we present a low-cost, low-power millimeter-wave (mmWave) skeletal joint localization system. High-quality point cloud data are generated using the self-developed BHYY_MMW6044 59–64 GHz mmWave radar device. A sliding window mechanism is introduced to extend the single-frame point cloud into multi-frame time-series data, enabling the full utilization of temporal information. This is combined with convolutional neural networks (CNNs) for spatial feature extraction and a bidirectional gated recurrent unit (BiGRU) for temporal modeling. The proposed spatio-temporal information fusion framework for multi-frame point cloud data fully exploits spatio-temporal features, effectively alleviates the sparsity issue of radar point clouds, and significantly enhances the accuracy and robustness of pose estimation. Experimental results demonstrate that the proposed system accurately detects 25 skeletal joints, particularly improving the positioning accuracy of fine joints, such as the wrist, thumb, and fingertip, highlighting its potential for widespread application in human–computer interaction, intelligent monitoring, and motion analysis.

## 1. Introduction

Human pose estimation is a critical task in computer vision, which aims to identify key joint positions in the human body based on sensor data to interpret human posture [1,2,3]. Accurate identification of these key points enables computers to interpret human behavior more effectively, which is essential in a variety of fields, such as health monitoring, motion analysis, human–computer interaction, and autonomous driving [4,5,6]. In autonomous driving, pose estimation aids vehicles in detecting pedestrian movements and predicting their behaviors for proactive decision-making. Similarly, in healthcare, it facilitates remote monitoring, particularly for evaluating the activities of elderly or recovering patients. Thus, human pose estimation has immense potential across diverse domains, particularly where remote and non-invasive sensing is needed.

Traditional methods for human pose estimation rely heavily on optical sensors, such as RGB and depth cameras, which can provide accurate results under optimal conditions [7,8]. However, these optical sensors depend on good lighting conditions, and their performance degrades significantly in low-light, overexposed, or occluded environments [9]. Additionally, they capture sensitive privacy information, including facial features, raising privacy concerns in scenarios such as home monitoring or medical applications. These limitations significantly reduce the applicability of optical methods in certain applications. Alternatively, wearable sensors (e.g., accelerometers, gyroscopes) can capture motion data and provide pose estimates under challenging conditions, such as in low light or occlusion [10,11,12]. However, prolonged use of such devices may cause discomfort, especially during frequent movement, as the sensors can slip or fall off, negatively impacting posture estimation accuracy. Additionally, these devices require regular maintenance and recharging, leading to a subpar long-term user experience. Use of WiFi signals for human pose estimation is a relatively new approach [13]. WIFI is very suitable for indoor scenarios due to its privacy protection and resistance to occlusion. However, its spatial resolution is low and its accuracy is limited by signal interference and multipath effects. Moreover, it relies on dense AP deployment and the channel state information processing is complicated.

Millimeter-wave (mmWave) radar, an emerging sensing technology with low-cost and low-power consumption, provides a non-invasive and privacy-friendly alternative for pose estimation [14,15,16]. By transmitting radio frequency (RF) signals and capturing their reflections, mmWave radar generates three-dimensional (3D) point clouds that can penetrate obstacles like smoke, dust, and clothing while maintaining stability in complex environments [17,18]. Unlike optical sensors, mmWave radar does not capture facial details, ensuring enhanced privacy protection. Furthermore, its compact antenna design and low power consumption make it highly suitable for portable devices and remote monitoring systems [19]. Despite these advantages, the sparse nature of mmWave radar point clouds presents challenges for accurate human pose estimation due to the limited number of reflection points [20,21,22].

To overcome the challenges associated with sparse point cloud data from mmWave radar, deep learning techniques have gained traction in recent years for improving pose estimation accuracy. These algorithms are adept at extracting meaningful features from large datasets, allowing them to infer detailed pose information from sparse radar point clouds. By leveraging deep learning, mmWave radar systems can effectively address the data sparsity problem, leading to more accurate and reliable pose estimation. With advancements in radar technology and deep learning algorithms, mmWave radar is poised to play a significant role in areas such as human–computer interaction, intelligent monitoring, and autonomous driving.

In this paper, we present a low-cost, low-power mmWave skeletal joint localization system, where motion data are collected using a self-developed mmWave radar device, and reference skeletal joint data are collected using a Microsoft Kinect V2 sensor. Unlike previous studies focused on point cloud-based activity recognition and localization, our proposed sliding window-based CNN-BiGRU architecture directly converts radar point cloud into 3D joint coordinates, enabling skeletal pose estimation without the need for video image support. For empirical evaluation, we used 150 min of exercise data from three volunteers of varying body sizes, covering 10 different movements and totaling 179,751 frames. The experimental results showed that the algorithm accurately estimated the 3D coordinates of 25 joints, with an average absolute error of only 2.72 cm, thereby confirming its validity.

The remainder of the paper is organized as follows: Section 2 provides a review of related work in the field of skeletal pose estimation. Relevant background theory for radar is presented in Section 3. Section 4 details the proposed method, including point cloud preprocessing, and the spatio-temporal information fusion network architecture based on CNN-BiGRU. Section 5 presents the experimental results and compares them with existing approaches in the literature. Finally, Section 6 concludes the paper.

## 2. Related Work

Human pose estimation methods can be categorized into three groups based on their working principles: model-based methods, feature-based methods, and machine learning-based methods. Model-based methods classify motions by constructing mathematical models or skeletal structures of the human body and comparing the similarity of detected targets to these models. These approaches typically rely on motion capture techniques or multi-view images to obtain highly accurate pose information. Lehment et al. [23] developed an upper body ellipsoid model composed of nine basic body modules, including the left and right upper arms, lower arms, hands, head, torso, and neck. This model can be represented as a 3D ellipsoid, which is easy to generate and manipulate. In the absence of color or texture information, the closest point to the input point cloud can be identified to compute the similarity of the likelihood function. Wu et al. [24] designed a human skeletal model with customized parameters based on T-pose. The model uses a depth image and a 3D point cloud as inputs, recursively finding other joint nodes with the torso center of mass as the root node; specifically, the information about the human skeleton is obtained by matching it with the model, which is able to accurately estimate the human body pose even in the presence of occlusions. Xu et al. [25] proposed a Gaussian Mixture Model-based reshaping method, which successfully realizes human shape estimation without requiring prior information by capturing the user’s pose and shape through multiple RGB-D sensors. The method is able to handle complex poses and occlusion situations, and realize different types of shape reshaping by automatically adjusting body attributes, such as height and weight. Lu et al. [26] introduced a novel unsupervised framework aimed at accurately learning 3D joint skeletons from a sequence of point sets captured by a single consumer-grade depth camera. This framework utilizes non-rigid point cloud alignment to establish point correspondences, generates skeletal structures through clustering, and performs precise joint learning using linear hybrid skinning.

Global features are a common feature-based approach with the advantages of low variance, simple computation, and intuitive representation. Commonly employed features include the color, texture, and shape of the human body. The human behavior tracking method proposed by Yuan et al. [27] is capable of automatically detecting motion feature points and correcting errors in detection through feature point trajectories. Specifically, the shoulder joint template is used to identify the shoulder joint position; the total geodesic distance from the shoulder point to the hand point is minimized using an iterative search method to locate the elbow joint. Additionally, a minimum distance constraint is introduced to predict the joint position in the subsequent frame to achieve accurate motion trajectory tracking. Hu et al. [28] proposed a method for curve skeleton extraction, which first converts multi-frame depth data captured by depth sensors into a point cloud. Then, the distance field and curvature are combined to capture the hybrid feature points. The associated faces and points are introduced to connect these hybrid feature points. Finally, a curve skeleton is generated. Patil et al. [29] proposed a pose tracking system based on multiple 3D light detection, ranging sensors and Inertial Measurement Unit (IMU) sensors. In this system, human skeletal parameters are estimated by extracting a point cloud from the LiDAR data, while the orientation of skeletal segments is captured using the IMU. The joint positions are then estimated, and the data from both sensors are fused in real time to correct for positional drift. Xu et al. [30] proposed a method to detect human joints using a single-frame point cloud captured by a time-of-flight (ToF) camera. This approach extracts 3D human contours from preprocessed data, then performs pose recognition with angle information and aspect ratio of contours. The positions of 14 joints are derived based on geometric features.

Model-based methods are constrained in dynamic scenes by the diversity and adaptability of models, which are usually sensitive to ambient lighting problems. Therefore, these methods are restricted to a few applications. Feature-based methods, on the other hand, may struggle with accurate feature extraction in the presence of complex backgrounds and occlusions, adversely affecting the final pose estimation. In light of these challenges, many researchers are now turning to deep learning-based approaches for human pose estimation, spurred by the rapid advancements in machine learning technology within the field of computer vision. Schnürer et al. [31] designed a real-time 3D pose estimation system using a single depth image. By employing CNNs to generate 2D confidence maps and incorporating depth information, the system predicts body joint positions while maintaining high frame rates and low resource usage. However, the depth mapping from 2D single-channel images does not represent an actual 3D representation. To address this limitation, Vasileiadis et al. [32] introduced a detection-based fully convolutional 3D-CNN architecture. This model leverages 3D voxel data to generate likelihood maps for each joint, extending the method to multi-person pose estimation through a bottom-up approach. As a non-contact RF sensor, mmWave radar offers effective protection of facial privacy and can be used as a complement or alternative to cameras in pose estimation tasks. Adib et al. [33] were the first to use an RF-capture system for human skeletal reconstruction, and output approximate human localization using FMCW signals in the 5.4 to 7.2 GHz frequency band, laying the foundation for further studies on skeletal pose estimation using millimeter waves. In 2018, researchers from the same group introduced RF-Pose, a system that localizes 14 body parts, including the head, neck, shoulders, elbows, wrists, hips, knees, and feet, using a Regional Proposal Network (RPN) followed by a CNN with a ResNet architecture [34]. In addition to localising only 14 joints, this work has not leveraged the ability of mmWave radar to generate high-quality point cloud. In 2020, Sengupta et al. [35] proposed mmPose, which enables the prediction of 17 skeletal joint coordinates by constructing a skeleton from point cloud using two IWR 1443 radar devices and a bifurcated CNN architecture. Subsequently, Sizhe et al. [36] proposed a mmWave-based assistive rehabilitation system called MARS to address movement disorders. *MARS* maps a 4D time-series point cloud to lower dimensions and reconstructs 19 human joints with their skeletons using a CNN. While the aforementioned studies are effective in accurately localizing the main joints of the human body, the precise localization of smaller and more granular radar cross-section joints, such as the wrists, thumbs and fingertips, remains a significant challenge.

In contrast to previous studies, our proposed human skeletal pose estimation system employs a self-developed BHYY_MMW6044 59–64 GHz mmWave radar device, combined with a unique CNN-BiGRU architecture for sliding window temporal processing and spatio-temporal feature modeling, which is capable of accurately predicting the 3D coordinates of all 25 skeletal joints of the human body. The main contributions of this paper are summarized as follows:

(1) Low-cost, low-power millimeter-wave skeletal joint positioning system: This system utilizes the self-developed BHYY_MMW6044 59–64 GHz mmWave radar device, which is characterized by low cost and low power consumption, making it suitable for large-scale deployment and prolonged operation. It is capable of generating high-quality millimeter-wave point cloud and providing accurate 3D spatial information, offering a stable and efficient solution for skeletal pose estimation, and ensuring high accuracy and robustness.

(2) Spatio-temporal information fusion framework based on multi-frame point cloud data: Unlike traditional methods that use single-frame point cloud input or multi-frame accumulation, we employ a sliding window mechanism to convert single-frame point cloud data into multi-frame time-series data. These frames are convolved individually, stacked in chronological order, and then input into the BiGRU module for time-series modeling. The sliding window technique effectively addresses the sparsity issue of radar point cloud by expanding the original data along the time axis. By combining the strengths of CNN and BiGRU, the model is better equipped to capture temporal dynamic changes and extract spatial features from each time window, significantly improving the accuracy and robustness of human pose estimation.

(3) Comprehensive skeletal joint detection: Our method successfully detects 25 skeletal joints, including wrist, thumb, and fingertip joints. This not only improves the accuracy of human pose estimation but also enhances the system’s applicability, accurately reflecting subtle movements and complex postures, thereby expanding its potential use in fields such as gesture control, precision manipulation, and interactive motion analysis.

## 3. Background Theory

Frequency Modulated Continuous Wave (FMCW) mmWave radar transmits millimeter-wave signals through the transmitting (TX) antenna, and utilizes the receiving (RX) antenna to acquire the echo signal reflected by the target. Positional information, such as range, velocity, and angle, is generated via a specific signal processing method [37,38,39]. The basic waveform of FMCW radar is a sawtooth wave, whose frequency varies linearly with time. Such a linear frequency-modulated pulse is generally referred to as a chirp, and each radar frame contains *N* chirps, as shown in Figure 1. In practice, FMCW radar usually transmits multiple frames at a time in order to obtain better results. Specifically, the start frequency is $f_{0}$, the time duration of a chirp is $T_{c}$, and the bandwidth is denoted as $B=kT_{c}$, where *k* is the sweep slope.

When the radar is relatively stationary with the detected target, the received echo signal is supposed to be at the same frequency as the transmitted signal. And the delay τ is only related to the distance *R*. Specifically, this relationship is expressed as τ=2R/c, where *c* is the speed of light. The transmitted signal and the echo signal pass through the mixer to obtain a differential frequency signal, also known as an intermediate frequency signal (IF). This IF signal can be expressed as(1)$$f\left(t\right)=Asin\left(2\pi f_{IF}t+\varphi_{0}\right)$$
where the amplitude of the IF signal is *A* and the frequency is $f_{IF}=k\tau =k2R/c$. The initial phase $\varphi_{0}$ is the phase difference between the transmit signal and the echo signal, so $\varphi_{0}=2\pi f_{0}\tau =4\pi R/\lambda$ (λ is the wavelength).

The frequency of the IF signal is proportional to the distance of the target. A fast Fourier transform (FFT), referred to as Range FFT, is performed in the digital domain to find the peak in the frequency domain and then the distance of the target is determined to be(2)$$R=\frac{cf_{IF}}{2k}$$

According to FFT theory, the range resolution is $R_{res}=c/2B$, which is only related to the chirp bandwidth *B*. Therefore, the range resolution of the FMCW radar is mainly improved by increasing the chirp bandwidth.

The velocity of the target cannot be calculated from a single chirp signal in the FMCW radar. Hence, the FMCW radar emits multiple chirps in one frame. For a moving target, each IF signal has the same frequency but a different phase. That is, the Range FFT results of IF signals from a moving object have the same peak location but differ in phase. The phase difference is $\Delta \varphi =4\pi \Delta R/\lambda =4\pi vT_{c}/\lambda$. Once the phase information of the target is obtained, the velocity of the target can be calculated as(3)$$v=\frac{\lambda \Delta \varphi }{4\pi T_{c}}$$

In order to detect multiple targets with different velocities at the same location, it is necessary to execute FFT on the chirp dimension, also known as a Doppler FFT. According to the theory of discrete Fourier transforms, the velocity resolution is $v_{res}=\lambda /2NT_{c}$.

The point cloud can be obtained from the range-Doppler heatmap generated after Doppler FFT, where each point reflects a pair of range and Doppler velocity information. To remove the environmental noise and non-target points, the constant false alarm detection method (CFAR) is usually adopted. Furthermore, the Smallest of Cell Average CFAR (SOCA-CFAR) algorithm is used in this work, because it has better performance than the traditional CFAR under multi-target conditions. Specifically, points with signal strength higher than a pre-defined threshold are retained in the point cloud, while the rest are discarded.

In order to obtain the three-dimensional spatial coordinates of the point cloud, it is also necessary to estimate the angular parameters. The angle of the target is usually estimated on the horizontal plane, which can also be referred to as the direction of arrival (DoA). From the aforementioned ranging and velocimetry methods, it can be seen that small changes in target distance can lead to a phase change in the IF signal at the peak of the Range FFT or Doppler FFT. The characteristics of the phase change at the peak can be used to estimate the target angle, and this method requires at least two RX antennas. An example of the RX antenna array is shown in Figure 2. When an echo signal from the target reaches adjacent RX antennas, it experiences a propagation distance difference. Given that the spacing between adjacent antennas is *d*, this distance difference is dsinθ, where θ represents the DoA of the signal. The phase change can be expressed as $\Delta \varphi =2\pi f_{0}\Delta t=2\pi \left(c/\lambda \right)\left(d\sin\theta /c\right)=2\pi d\sin\theta /\lambda$. Then, θ can be calculated as(4)$$\theta =\sin^{-1}\left(\frac{\lambda \Delta \varphi }{2\pi d}\right)$$

This phase change Δφ is essential for angle estimation. By performing the third FFT along the antenna dimension, also known as Angle FFT, Δφ is reflected in the peak. Angle resolution is $\theta_{res}=\lambda /dM\cos\theta$, which is related to the number of RX antennas (*M*). The larger *M*, the better the resolution. If d=λ/2, the minimum angle resolution is $\theta_{res}=2/M$.

Since the antenna array of the BHYY_MMW6044 59–64 GHz radar is distributed in both horizontal and vertical directions, it enables estimation of both the azimuth and elevation angles of the target. Specifically, an Angle FFT is performed along the horizontal antenna dimension to determine the target’s azimuth angle $\theta_{azi}$, while another Angle FFT is applied along the vertical antenna dimension to obtain the elevation angle $\theta_{ele}$ of the target.

With the mmWave radar signal processing method described above, a target point cloud can be detected from the environment and information such as the distance *R*, Doppler velocity *D*, azimuth angle $\theta_{azi}$, and elevation angle $\theta_{ele}$ of the target can be obtained. For each point, a spherical coordinate system is transformed to a 3D Cartesian coordinate system using the following formula:(5)$$\begin{array}{cc} \; & x=R\sin\theta_{ele}\cos\theta_{azi}\\ & y=R\sin\theta_{ele}\sin\theta_{azi}\\ & z=R\cos\theta_{ele} \end{array}$$
where x,y,z denote 3D Cartesian coordinates. Together with the 3D coordinates and Doppler velocity *D*, a 4D point cloud is formed for further processing later.

The radar parameters used in this study are shown in Table 1. The self-developed BHYY_MMW6044 59–64 GHz mmWave radar is configured with four TX antennas and four RX antennas. Since its antenna array is distributed both horizontally and vertically, it is equivalent to eight RX antennas in the horizontal direction and three RX antennas in the vertical direction. Thus, the azimuth and elevation resolution are 14.32° and 20°, respectively. According to the waveform parameters in the table, the range resolution is 5.00 cm and the velocity resolution is 0.13 m/s.

## 4. Proposed Approach

This section presents a human skeletal pose estimation system that utilizes mmWave radar and a CNN-BiGRU architecture. The system accurately estimates the 3D coordinates of 25 skeletal joints of the human body by processing raw mmWave point cloud data, as shown in Figure 3. The process involves the following steps, depicted in Figure 4: (1) Acquisition of high-quality point cloud data using the BHYY_MMW6044 radar device; (2) Temporal processing of multi-frame point cloud data using the sliding window technique to construct a sequence of consecutive frames with a fixed step size; and (3) Spatio-temporal feature modeling using the CNN-BiGRU architecture to accurately predict the 3D coordinates of the 25 skeletal joints.

### 4.1. Point Cloud Preprocessing

#### 4.1.1. Input Data Reshaping

The system processes input sequences of 4D radar point clouds, where each point contains four features: x-, y-, and z-coordinates (x, y, z), and Doppler velocity D. In this work, the radar captures the first 64 reflection points to form a data frame. If fewer than 64 points are detected, the remaining part of the frame is padded with zeros to ensure a consistent input frame size of 64×4. The reflected chirp signals reach the radar in a random sequence due to constant changes in human pose and uncertainties in round-trip delay, resulting in a random order of points in each frame. While CNNs are generally robust to translation, scaling, and rotation, this randomness increases the difficulty of model learning. To address this problem, we sort the points in each frame by their x-, y-, and z-coordinates in ascending order. Specifically, points are first sorted by the x-coordinate, followed by the y-coordinate if the x-values are identical, and by the z-coordinate if both the x and y values are the same. This sorting does not change the relative distances between points while only adjusting their input order for the model. The ordered data help the convolutional kernel capture spatial patterns more efficiently, improving the accuracy and effectiveness of feature extraction.

After sorting the data, the dimension of the input features is 64×4, where each frame contains 64 data points, each with four features. To optimize the input for the CNN model, we reorganize these 64 points into an 8×8 matrix in row-first order to make its structure closer to the traditional image data format. This transformation also helps the CNN to extract spatial features more efficiently, and the final input dimension is 8×8×4.

#### 4.1.2. Multi-Frame Point Cloud Temporal Modeling Based on Sliding Window

In mmWave radar human pose estimation tasks, traditional methods typically use single-frame point cloud input or multi-frame cumulative processing. Due to the high noise, dynamic variations and sparsity of radar point cloud data, single-frame processing often struggles to accurately capture the continuous features of human motion. Although multi-frame accumulation can alleviate this issue, it may introduce motion blur, which affects the accuracy of pose estimation. To address this, we introduce a sliding window mechanism, where each frame of 8×8×4 point cloud data is transformed into a sequence of time windows of length step (with the shape of step×8×8×4), as shown in Figure 5. In this way, the data at each time point consist of multiple consecutive time frames, which not only retains the time dimension information but also captures short-term dynamic features within the local time range, reduces the interference of single-frame noise on pose estimation, and avoids the motion blurring problem caused by multi-frame accumulation. Additionally, the sliding window mechanism effectively mitigates the sparsity issue of radar point clouds by expanding the raw data along the time axis. The window size steps of the sliding window are experimentally optimized, with the optimization process balancing computational efficiency and detection accuracy. The specific optimization details and ablation experimental results are provided in Section 5.3.2.

### 4.2. Spatio-Temporal Information Fusion Network Architecture Based on CNN-BiGRU

The CNN-BiGRU-based spatio-temporal information fusion network architecture is designed to achieve high-precision human pose estimation using mmWave radar. It primarily consists of a CNN module, a BiGRU module, and a fully connected layer, as shown in Figure 6. First, the input point cloud data are processed using a sliding window to generate multi-frame time-series data, which are then fed into the CNN module for spatial feature extraction. The extracted spatial features are subsequently passed to the BiGRU module in chronological order for temporal modeling, capturing the temporal dependencies between the features. By effectively fusing spatio-temporal information, the model can accurately track dynamic changes in human motion, enhancing the accuracy and robustness of pose estimation. Finally, the spatio-temporal features are passed through the fully connected layer to output the 3D coordinates of 25 human skeletal joints.

The input to the CNN module consists of step×8×8×4 point cloud data after sliding window processing. The convolution operation is applied separately to each frame to extract spatial features. The module comprises two CNN layers with a kernel size of 3×3, and the number of channels in these layers is 16 and 32, respectively. The activation function used is the Rectified Linear Unit (ReLU). The momentum of the batch normalization layer is set to 0.95 to stabilize the data distribution of each layer, reduce internal covariate shift by normalizing the data in each mini-batch, and accelerate the training process, thereby improving the model’s convergence speed. To prevent overfitting, a Dropout layer (with a dropout rate of 0.3) is added after each CNN layer, enhancing the model’s generalization capability. Finally, the outputs from the CNN layers are passed through a flattening layer, which transforms the 2D feature maps into 1D vectors. These vectors are arranged in chronological order to create time-series data that meet the input requirements of the BiGRU module, providing sufficient spatial feature information for subsequent temporal modeling.

The BiGRU network consists of 128 neurons and uses ReLU as the activation function. With its bi-directional GRU structure, the model can capture both forward and backward information in the time series, effectively modeling the long- and short-term temporal dependencies in human pose estimation. This is crucial for handling dynamic changes and complex temporal correlations in human movements. Compared to the Long Short-Term Memory (LSTM) network, the GRU has fewer parameters and higher computational efficiency, which reduces training time while maintaining performance, making it well suited for real-time pose estimation tasks. The BiGRU layer is followed by a flattening layer, and its output is processed by the Batch Normalization layer and the Dropout layer, with parameters set to 0.95 and 0.4, respectively. Finally, the output of the BiGRU module is passed to the fully connected layer for further processing and output.

The fully connected layer comprises a two-layer network. The first fully connected layer contains 128 neurons and employs ReLU as the activation function. The output layer consists of 75 neurons, corresponding to the x-, y-, and z-coordinates of 25 skeletal joints, and uses linear activation. Given the strong spatial correlation between the radar point cloud and the coordinates of actual human skeletal joints, the loss function is defined as the Mean Squared Error (MSE) between the predicted joint positions and their actual positions on the ground.(6)$$L=\frac{\sum_{j=1}^{J}{\left(x_{i}-{\hat{x}}_{i}\right)}^{2}+\sum_{j=1}^{J}{\left(y_{i}-{\hat{y}}_{i}\right)}^{2}+\sum_{j=1}^{J}{\left(z_{i}-{\hat{z}}_{i}\right)}^{2}}{3J}$$
where $x_{i}$, $y_{i}$, $z_{i}$ represent the ground truth coordinates; ${\hat{x}}_{i}$, ${\hat{y}}_{i}$, ${\hat{z}}_{i}$ denote the estimated coordinates of the skeletal joints; and *J* is the number of skeletal joints.

## 5. Experimental Evaluations

### 5.1. Experimental Setup and Dataset

**mmWave radar:** We developed a mmWave radar, named BHYY_MMW6044, and its structure is shown in Figure 7. Its RF chip is RC6011 and the processing chip is SKY32B750, both of which are independently developed by SKYRELAY (Beijing) Technology Co., Ltd., Beijing, China. The RC6011 packs four transmit and four receive channels, operating within the 60 GHz frequency band. Meanwhile, the SKY32B750 emerges as a dedicated radar processing chip, armed with advanced radar computing cores such as FFT and CFAR, which reduces the complexity, cost, and power consumption of the radar system. In the current study, all radar processing tasks are executed on BHYY_MMW6044. Subsequently, the point cloud results are transmitted in real-time to a computer via the Universal Asynchronous Receiver–Transmitter (UART) interface. The average power of BHYY_MMW6044 is only 570 mW.

**Kinect V2 sensor:** We acquire the ground truth reference by employing Microsoft Kinect V2. As depicted in Figure 8, the Kinect is set at a height of 1 m, while the radar is positioned 2 m high, and their horizontal alignments are roughly equivalent. The Kinect V2 sensor connects to the computer through a USB port and captures images at a sampling rate of 30 Hz. Subsequently, the computer processes these images to derive the 3D coordinates of 25 human joints depicted in Figure 3. These coordinates serve as key reference points for labeling and testing during the training stage.

**Test datasets:** We constructed a large-scale dataset to validate the effectiveness of the sliding window-based CNN-BiGRU architecture. The experiment involved three subjects of varying body sizes, each performing ten predefined movements: left upper limb stretch, right upper limb stretch, double upper limb stretch, left front lunge, right front lunge, squat, left lunge, right lunge, left limb stretch, and right limb stretch, involving varying degrees of joint movement and postural changes. This diverse set of actions enables a comprehensive evaluation of the model’s ability to capture complex human postures, while also enhancing its robustness and generalization ability in real-world application scenarios. Data acquisition was conducted in a controlled experimental environment, as shown in Figure 9, with each subject performing each action for 5 min. After data collection, we synchronized the timestamps of the Kinect and mmWave radar to ensure accurate pairing of the frames, resulting in 179,751 frames of aligned data. Each frame of the Kinect data contained 25 3D joint coordinates, while the radar point data consisted of four dimensions: the 3D spatial coordinates and Doppler velocities. This dataset serves as the foundation for subsequent model training and analysis.

**CNN-BiGRU training details:** The dataset was divided chronologically into a training set (60%, 107,836 frames), a validation set (20%, 35,955 frames), and a test set (20%, 35,960 frames). The CNN-BiGRU architecture was implemented in Python, using Keras 2.7.0 and Tensorflow 2.7.0. Training was conducted on a system with an Intel(R) Core(TM) i5-14600KF CPU and GeForce RTX 2080 Ti GPU (Intel Corporation; Santa Clara, CA, USA). The model was trained using the Adam optimizer with a batch size of 128 for 150 epochs, an initial learning rate of 0.001, and step set to 12.

### 5.2. Accuracy of 3D Skeletal Joint Position Estimation

Human pose estimation datasets are often specific to the sensor type, data format, and acquisition conditions, which can significantly impact model performance and applicability. To ensure a fair comparison between different models, it is essential to select a consistent dataset for comparative experiments. In this study, we selected both the self-constructed dataset and the publicly available dataset provided in [36], using *MARS* [36] and *mm-Pose* [35] as the benchmark models for performance comparison. The *MARS* model uses CNNs to extract 5D radar point cloud features and outputs the coordinates of 19 skeletal joints, while the *mm-Pose* model is based on a forked CNNs architecture and predicts the locations of 17 skeletal joints. For a unified comparison, we adjusted the output layer parameters of both models to enable them to output the coordinates of 25 skeletal joints. Model performance was evaluated using the mean absolute error (MAE) and the root mean square error (RMSE) metrics. To reduce systematic errors, each model was trained 10 times, and the results were averaged. Figure 10 and Figure 11 show the localization errors of the three models on the *MARS* dataset for the 25 skeletal joints, while Figure 12 and Figure 13 show the corresponding errors on the self-built dataset, and Figure 14 displays the visualization results of our method. Table 2 presents the mean errors of the three models across the two datasets.

Experimental results based on both the self-built and the *MARS* datasets demonstrate that our model significantly outperforms the *MARS* and *mm-Pose* models in two key metrics: MAE and RMSE. On the self-built dataset, for the X-, Y-, and Z-axis coordinates of the 25 skeletal joints, our model achieved an average MAE that was 47.59% and 41.88% lower than that of *MARS* and *mm-Pose*, respectively. Additionally, the RMSE was reduced by 39.04% and 38.44%, compared to *MARS* and *mm-Pose*, respectively. A similar improvement in performance was observed on the *MARS* dataset. This enhancement is primarily attributed to the optimization of our model’s spatio-temporal information fusion approach. Unlike the *MARS* and *mm-Pose* models, which rely on single-frame inputs and simple CNN structures, our model applies frame-by-frame convolution on multi-frame point cloud data through a sliding window mechanism and leverages BiGRU for temporal modeling. This fully exploits the temporal features and efficiently captures the dynamics across consecutive time steps, leading to a significant boost in human pose estimation accuracy. Notably, for fine joint positioning (e.g., wrist and fingertips), our model demonstrates clear advantages in terms of higher accuracy, stability, and robustness.

### 5.3. Ablation Study

#### 5.3.1. Effect of Radar Point Cloud Features

To explore the effect of different radar point cloud features on the performance of the human pose estimation model, four different input feature configurations were designed for comparison:Configuration-1: 3D point cloud (containing only the spatial coordinates *x*, *y*, *z*).Configuration-2: 4D point cloud (containing the spatial coordinates *x*, *y*, *z*, and Doppler velocity *v*).Configuration-3: 4D point cloud (containing the spatial coordinates *x*, *y*, *z*, and intensity *I*).Configuration-4: 5D point cloud (containing the spatial coordinates *x*, *y*, *z*, Doppler velocity *v*, and intensity *I*).

It is worth noting that the intensity information is normalized to ensure it maintains a similar scale to the other features, thereby avoiding bias in the model training process due to differences in feature scales. By incrementally increasing the dimensionality of the point cloud features, we aim to explore the contribution of different features to pose estimation performance and assess whether the addition of additional features (e.g., velocity and intensity) can further improve model accuracy. All other experimental conditions were kept consistent to ensure a thorough analysis of the impact of varying point cloud dimensional configurations on model performance.

The average MAE and RMSE for 25 human skeletal joints on the X-, Y-, and Z-axes for these four configurations are presented in Table 3. The experimental results indicate a significant difference in the performance of various point cloud feature configurations for human pose estimation. The 4D (*x*, *y*, *z*, *v*) configuration performed the best, achieving the highest accuracy. This suggests that the inclusion of velocity features aids in capturing the dynamic changes of human joints, particularly across consecutive time steps, and allows for more accurate prediction of joint displacements, thereby improving pose estimation accuracy. In contrast, the 4D (*x*, *y*, *z*, *I*) and 5D configurations underperformed, yielding lower results than both the 4D (*x*, *y*, *z*, *v*) and 3D configurations. This implies that the intensity feature did not provide the expected performance enhancement. The intensity feature in radar measurements is influenced by factors such as the surface characteristics of the reflected object and the launch angle, leading to inconsistency and low correlation with joint positions. As a result, the model struggles to effectively utilize this feature, negatively impacting the stability and accuracy of pose estimation. Taking all these considerations into account, we selected the 4D (*x*, *y*, *z*, *v*) configuration as the final input feature to achieve the best pose estimation.

#### 5.3.2. Effect of Sliding Window Length Step

To improve the accuracy of the human pose estimation model, we introduce a sliding window mechanism that transforms each frame of 8×8×4 point cloud data into a time window sequence of length step. This approach enables the model to capture short-term dynamic features within a local time range, compensating for the lack of temporal information in single-frame data and avoiding motion blurring that may arise from accumulating multiple frames. In this section, we focus on the necessity of incorporating the sliding window mechanism to convert single-frame point cloud input into multi-frame input, and analyze the impact of different sliding window lengths (step) on pose estimation results. Figure 15 illustrates the average errors of 25 human skeletal joints on the X-, Y-, and Z-axes for various step values, where step=1 indicates that a single-frame point cloud is used as input.

As shown in Figure 15, the prediction error of the model is largest when step=1, indicating that single-frame point cloud input fails to effectively capture the temporal characteristics of human body movement, leading to significant estimation errors. As step increases, the accuracy of skeletal joint prediction gradually improves and stabilizes. This result demonstrates that the introduction of the sliding window mechanism allows the model to utilize continuous time-step information, thereby better capturing the dynamic changes in human pose and significantly enhancing pose estimation accuracy. As step increases, the model incorporates more temporal information, further improving prediction accuracy. However, when step becomes too large, more frames need to be processed during training, which may increase the computational burden. Therefore, an optimal balance between accuracy and efficiency must be found. Ultimately, we selected step=12 as the optimal configuration.

#### 5.3.3. Effect of CNN Module

The introduction of a batch normalization layer in the CNN module can accelerate the training process and enhance model stability. The max-pooling layer, a commonly used hierarchical structure in CNNs, effectively reduces model complexity and helps prevent overfitting. However, given the small amount of input radar point cloud data, using the max-pooling layer may result in the loss of local information, thereby affecting prediction accuracy. To verify the effectiveness of introducing a batch normalization layer and excluding the max-pooling layer in the CNN module, we designed four different model configurations for comparison:Without Batch Normalization and Max-Pooling Layers: this model serves as the baseline and is used to evaluate performance without any additional optimization.Adding a Batch Normalization Layer After the Convolutional Layer: this configuration accelerates training and improves model stability by introducing a batch normalization layer after the convolutional layer.Introducing a Max-Pooling Layer After the Convolutional Layer: a max-pooling layer is added to reduce model complexity and prevent overfitting, with its impact on performance evaluated.Applying Both Batch Normalization and Max-Pooling Layers After the Convolutional Layer: this configuration combines both batch normalization and max-pooling layers to test the effect of their combination.

The average MAE and RMSE of 25 human skeletal joints on the X-, Y-, and Z-axes of the four models are presented in the Table 4.

As shown in Table 4, the configuration with the addition of a batch normalization layer performs the best, while the configuration with both a batch normalization layer and a max-pooling layer after the convolutional layer performs the worst. This suggests that the batch normalization layer is effective in mitigating internal covariate shift, enhancing the stability and training efficiency of the model, and significantly improving pose estimation accuracy. In contrast, the introduction of the max-pooling layer is less effective. Since the task involves mapping the mmWave point cloud to the 3D coordinates of skeletal joints, the max-pooling operation may overemphasize local maximum features, resulting in the loss of important spatial information. Particularly in human pose estimation, accurate joint position prediction relies on fine-grained spatial features. As such, the pooling operation may weaken the model’s ability to capture these critical features, which negatively impacts prediction accuracy. Therefore, based on the experimental results, we choose to retain only the batch normalization layer and exclude the max-pooling layer in the CNN module to ensure optimal performance in human pose estimation.

#### 5.3.4. Effect of BiGRU Module

In this section, we compare various model configurations to validate the advantages of the BiGRU module in capturing the temporal features of human motion and its contribution to human pose estimation accuracy. The configurations are as follows:Removal of the BiGRU Module: only the input data are processed using CNNs, eliminating the temporal modeling component. This configuration evaluates the impact of temporal information on model performance.Replacement of BiGRU with GRU: the bidirectional GRU is replaced with a unidirectional GRU to assess the role of the bidirectional structure in capturing temporal features.Replacement of BiGRU with a Multilayer Perceptron (MLP): the BiGRU is substituted with an MLP to test the performance of fully connected networks in temporal modeling.Replacement of BiGRU with a Transformer: This setup employs the Transformer architecture to replace the BiGRU, exploring the role of the self-attention mechanism in time-series feature extraction.Retention of the BiGRU Module: This is our original model, which contains both CNN and BiGRU components, used here for comparative evaluation.

The average MAE and RMSE for 25 human skeletal joints on the X-, Y-, and Z-axes of the four models are shown in Table 5.

The experimental results demonstrate that excluding the BiGRU module leads to a significant degradation in the model’s performance on the pose estimation task, with an increase in the average positioning error. In contrast, our model exhibits optimal performance, underscoring the importance of the BiGRU module in time-series modeling. The GRU, due to its unidirectional structure, is unable to effectively utilize information from both the preceding and following contexts in the time series, resulting in a lack of comprehensive temporal dependence and lower estimation accuracy. The MLP, as a fully connected network, lacks temporal modeling capability and fails to handle temporal dependencies effectively, which impedes its ability to capture dynamic joint movements. The Transformer, while performing well on large datasets, incurs a significant computational overhead on smaller datasets, limiting its efficiency and failing to demonstrate advantages over BiGRU in handling pose changes. In tasks such as action recognition or pose estimation, future actions or changes have a significant impact on the current pose. BiGRU can process both past and future-oriented data to better model the dynamic changes in human pose, thereby significantly improving the accuracy and robustness of pose estimation, particularly in the localization of complex actions and subtle joints.

### 5.4. Power and Execution Time Analysis

To evaluate the computational efficiency of the model, we calculated its floating-point operations (FLOPs) using the keras-flops tool. For an input sequence with a sliding window size of step=12, the model requires approximately 8.14 MFLOPs (8.14 million operations) for a single inference. Of these, the BiGRU layer accounts for about 63% of the total operations, while the CNNs accounts for about 32%. This indicates that the computational load is primarily concentrated in the spatio-temporal feature fusion component. To further validate the real-time performance of the model, we tested its inference capabilities on an NVIDIA RTX 2080 Ti GPU. For the test dataset containing 35,960 frames, the total inference time was 6012.12 ms, with a total power consumption of 165 W. The average single-frame inference time was 0.167 ms/frame, and the energy consumption per frame was approximately 0.0276 J. These results demonstrate that our model completes inference with very low latency at each time step, fully meeting real-time requirements. Consequently, it is well suited for real-time human pose estimation, dynamic tracking, and other tasks requiring high computational efficiency.

### 5.5. Limitations

While significant progress has been made in this research, several limitations remain. The designed movement patterns are relatively simple, and as a result, the reliability of joint output may be compromised when the human body performs more complex actions, such as turning around. Furthermore, the current design ensures that all 25 joint points remain visible without occlusion, but challenges arise when body joints are obscured. Future work should focus on addressing the issue of joint occlusion. However, with further training that incorporates a broader range of behavioral data and optimization of the network architecture, the potential of mmWave radar for human joint estimation remains highly promising. It is anticipated that this capability will be enhanced, potentially enabling the simultaneous tracking of multiple human skeletal postures. Such advancements will lead to more accurate and robust estimations, offering new opportunities for applications in various fields, including human–computer interaction, intelligent monitoring systems, and advanced sports analytics.

## 6. Conclusions

We developed a low-cost, low-power mmWave skeletal joint localization system that utilizes the self-developed BHYY_MMW6044 59–64 GHz millimeter-wave radar device to generate high-quality point cloud data and construct multi-frame time-series data through a sliding window mechanism. Spatial features are extracted by combining with CNNs, while temporal dependencies are modeled by BiGRU, facilitating the fusion of temporal and spatial information. This approach effectively alleviates the sparsity issue of mmWave point clouds and significantly improves the accuracy and robustness of pose estimation. Experimental results show that our method outperforms the existing *MARS* and *mm-Pose* models on both the self-constructed dataset and the public dataset. It exhibits high accuracy in detecting 25 skeletal joints, with significant advantages, particularly in the localization of fine joints, such as the wrists, thumbs, and fingertips. Ablation experiments further validate the effects of the sliding window mechanism, BiGRU module, and different point cloud features on model performance, demonstrating the effectiveness of the proposed framework.

Despite the significant results achieved in this study for mmWave radar-based human pose estimation, the quality of the point cloud may degrade in the presence of occlusion, which, in turn, affects pose estimation accuracy. Future research could focus on optimizing point cloud completion strategies, particularly in occlusion scenarios. Additionally, augmented modeling of spatio-temporal features using attention mechanisms or graph neural networks could improve the robustness of the model. Furthermore, combining multi-sensor fusion techniques (e.g., inertial measurement units) to handle more complex pose estimation tasks, such as large motion, dramatic attitude changes, and multi-person interaction scenarios, will further enhance the system’s adaptability and applicability across diverse domains.

## Figures and Tables

**Figure 1 sensors-25-01070-f001:**
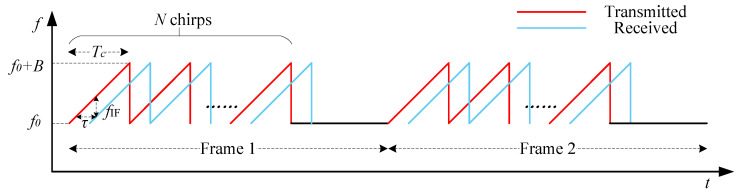
FMCW radar transmit and receive waveforms.

**Figure 2 sensors-25-01070-f002:**
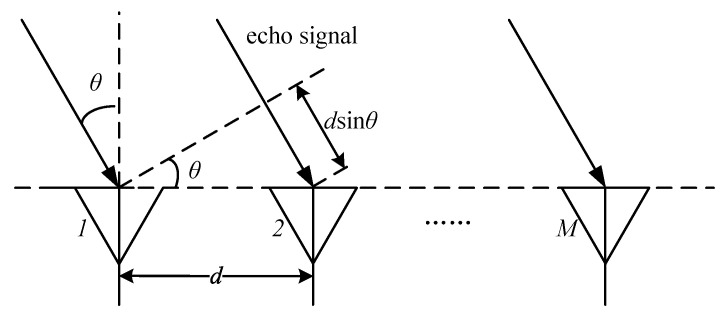
FMCW radar RX antenna array and phase relationship.

**Figure 3 sensors-25-01070-f003:**
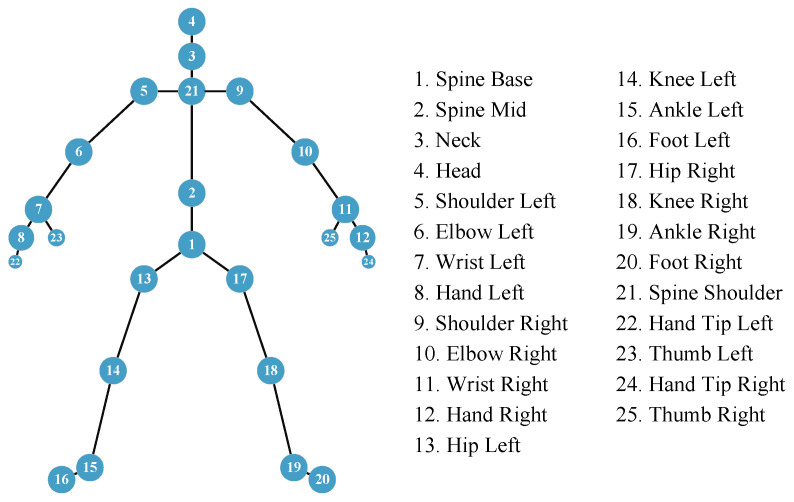
List and locations of 25 skeletal points.

**Figure 4 sensors-25-01070-f004:**
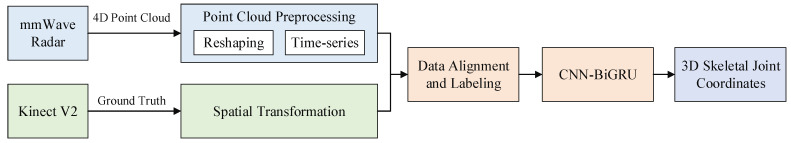
Overall flowchart of the human skeletal pose estimation system based on mmWave wave radar and CNN-BiGRU.

**Figure 5 sensors-25-01070-f005:**
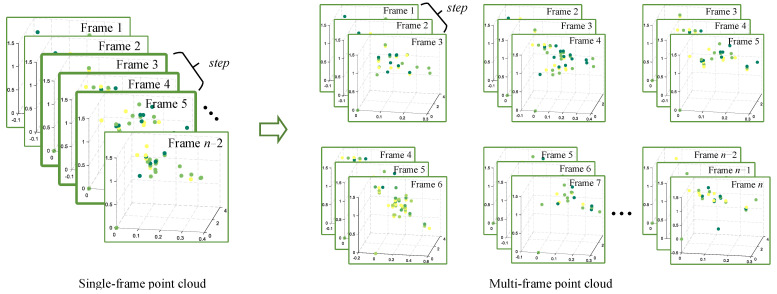
Multi-frame point cloud temporal modeling based on sliding windows.

**Figure 6 sensors-25-01070-f006:**
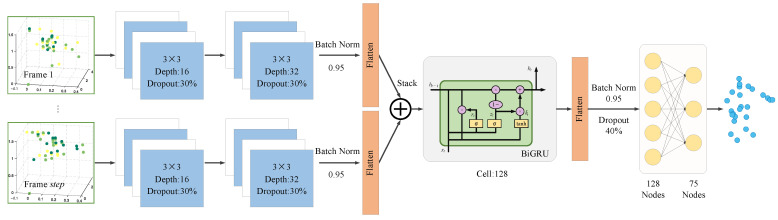
Spatio-temporal information fusion network architecture based on CNN-BiGRU.

**Figure 7 sensors-25-01070-f007:**
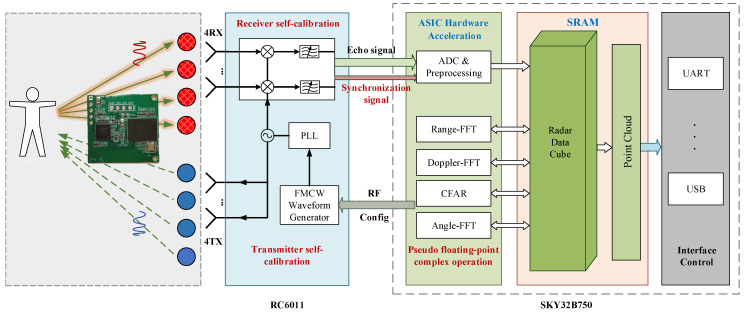
mmWave radar structure.

**Figure 8 sensors-25-01070-f008:**
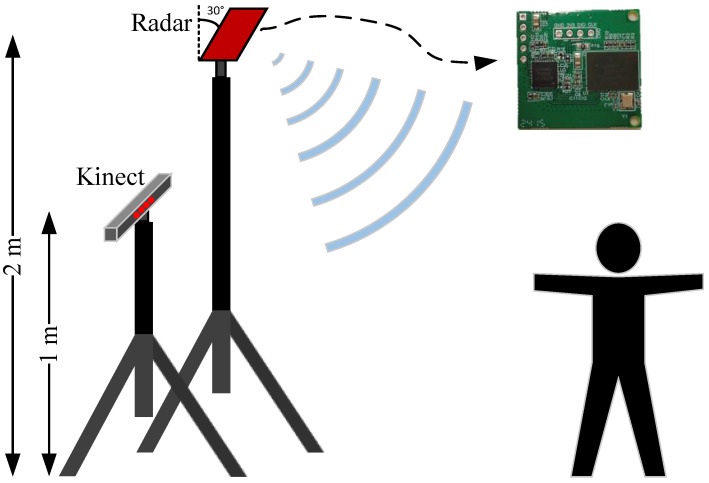
Experimental setup with one radar and one Kinect.

**Figure 9 sensors-25-01070-f009:**
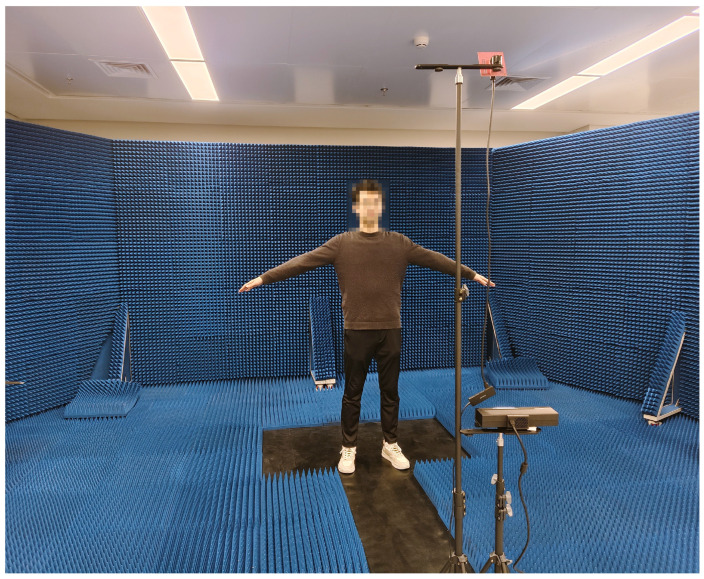
Experimental environment.

**Figure 10 sensors-25-01070-f010:**
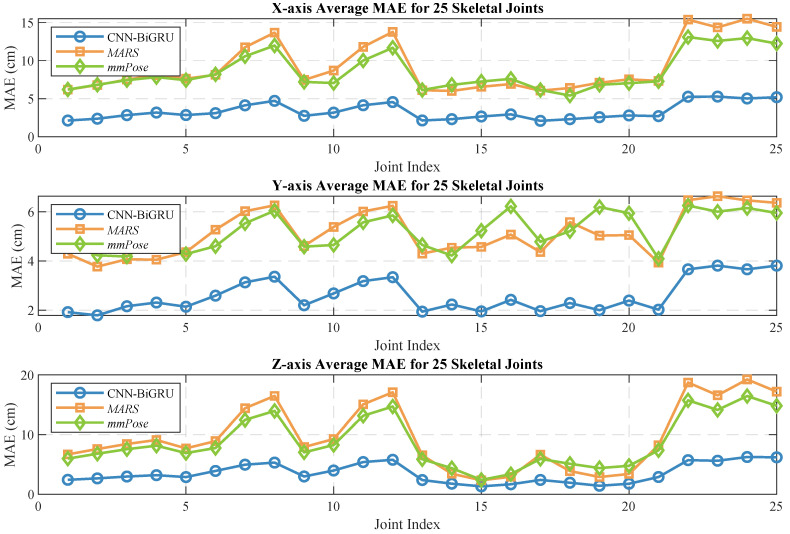
Average MAE for 25 human skeletal joints (*MARS* dataset).

**Figure 11 sensors-25-01070-f011:**
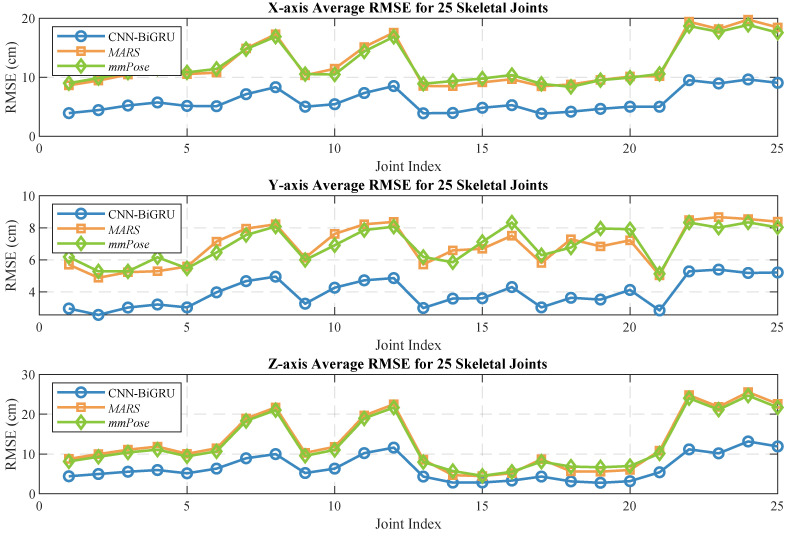
Average RMSE for 25 human skeletal joints (*MARS* dataset).

**Figure 12 sensors-25-01070-f012:**
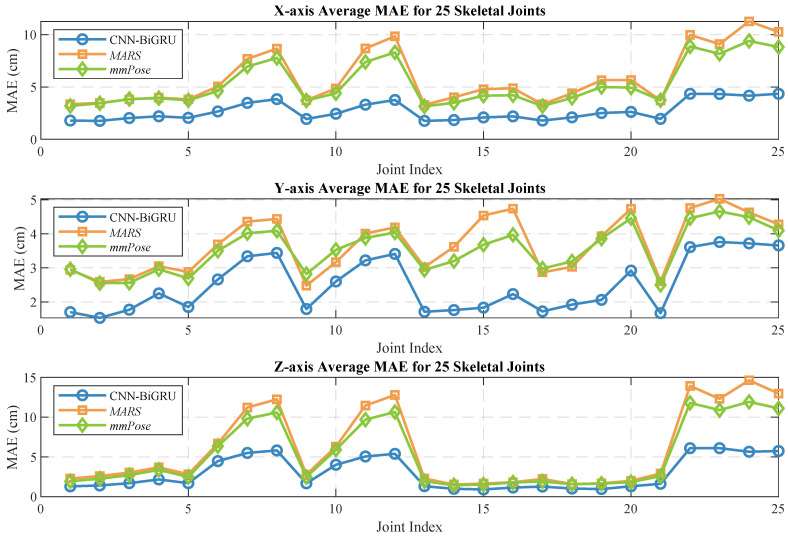
Average MAE for 25 human skeletal joints (self-built dataset).

**Figure 13 sensors-25-01070-f013:**
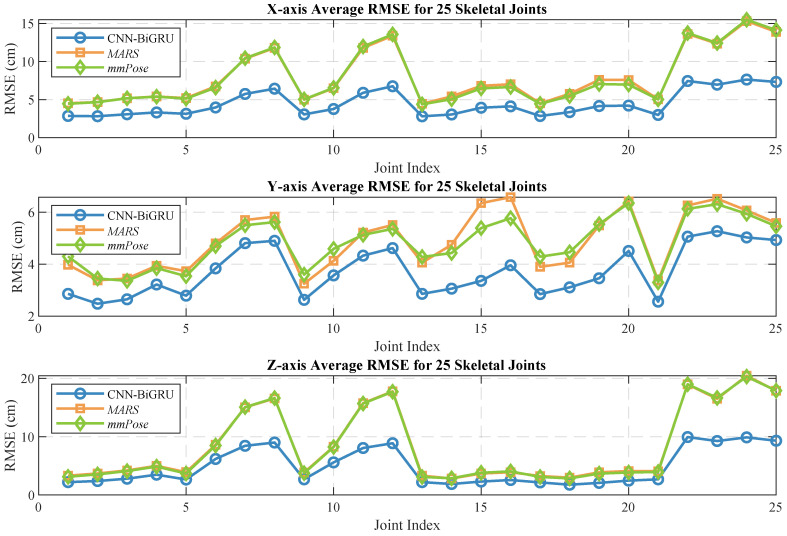
Average RMSE for 25 human skeletal joints (self-built dataset).

**Figure 14 sensors-25-01070-f014:**
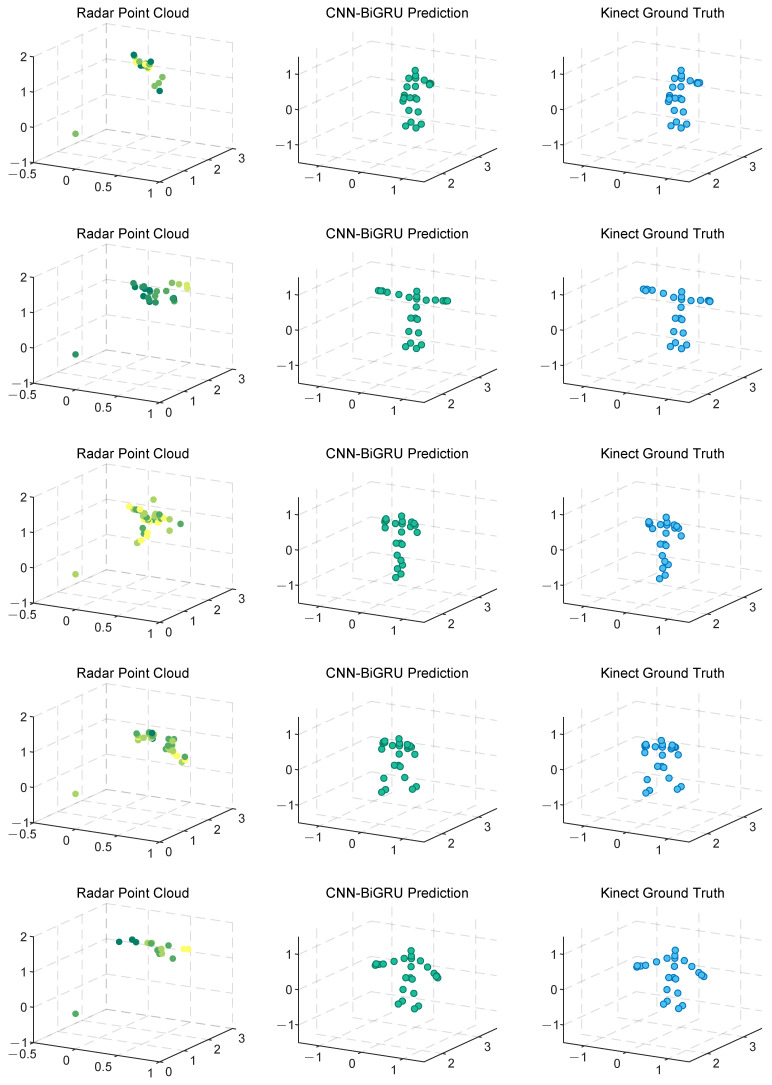
Demonstration of CNN-BiGRU reconstructing human skeletal joints from point cloud. From left to right, it shows radar point cloud, CNN-BiGRU prediction, and ground truth, respectively. The movements from top to bottom are left upper limb stretch, double upper limb stretch, left front lunge, right front lunge, and left lunge (self-built dataset).

**Figure 15 sensors-25-01070-f015:**
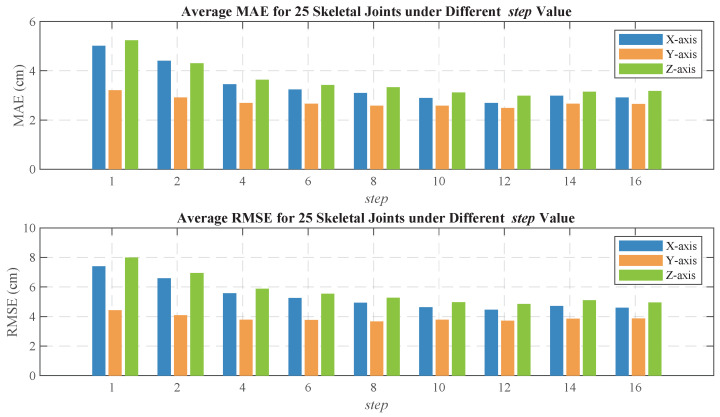
Average localization error for 25 human skeletal joints under different step value.

**Table 1 sensors-25-01070-t001:** Key parameters of self-developed BHYY_MMW6044 59–64 GHz mmWave radar device.

Symbol	Description	Value	Symbol	Description	Value
$f_{0}$	Starting frequency	59.5 GHz	$R_{res}$	Range resolution	5.00 cm
*B*	Bandwidth	3.22 GHz	$v_{res}$	Velocity resolution	0.13 m/s
$T_{c}$	Chirp duration	80 μs	$\theta_{a.res}$	Azimuth resolution	$14.32^{{}^{\circ}}$
$T_{frame}$	Frame duration	50 ms	$\theta_{e.res}$	Elevation resolution	$38.20^{{}^{\circ}}$
$N_{TX}$	No. of TX antennas	4	$N_{RX}$	No. of RX antennas	4
*N*	No. of chirps per frame	64	$f_{s}$	Sampling frequency	10 Msps

**Table 2 sensors-25-01070-t002:** Average localization error for 25 human skeletal joints across different models.

Dataset	Model	*x* (cm)		*y* (cm)		*z* (cm)		Average (cm)	
		MAE	RMSE	MAE	RMSE	MAE	RMSE	MAE	RMSE
*MARS*	*MARS*	9.25	12.26	5.15	6.92	9.63	12.91	8.01	10.70
	*mmPose*	8.55	12.24	5.20	6.94	8.69	12.55	7.48	10.58
	Ours	3.32	5.96	2.60	3.93	3.51	6.54	3.14	5.47
Self-built	*MARS*	5.90	7.99	3.69	4.89	5.98	8.47	5.19	7.12
	*mmPose*	5.30	7.92	3.52	4.83	5.21	8.40	4.68	7.05
	Ours	2.69	4.46	2.49	3.71	2.98	4.84	2.72	4.34

**Table 3 sensors-25-01070-t003:** Average localization errors of 25 human skeletal joints for models trained using different point cloud features.

Features	*x* (cm)		*y* (cm)		*z* (cm)		Average (cm)	
	MAE	RMSE	MAE	RMSE	MAE	RMSE	MAE	RMSE
xyz	3.12	4.92	2.94	4.23	3.33	5.24	3.13	4.79
xyzv	2.69	4.46	2.49	3.71	2.98	4.84	2.72	4.34
xyzI	4.08	6.68	3.23	4.71	4.34	7.22	3.88	6.20
xyzvI	4.20	6.68	3.24	4.65	4.20	6.83	3.88	6.05

**Table 4 sensors-25-01070-t004:** Average localization errors of 25 human skeletal joints for models trained using different CNN module. (BN denotes batch normalization layer, MP denotes max-pooling layer, ✓ indicates that this layer was used, × indicates that this layer was not used).

Module	*x* (cm)		*y* (cm)		*z* (cm)		Average (cm)	
	MAE	RMSE	MAE	RMSE	MAE	RMSE	MAE	RMSE
BN× MP×	2.94	4.96	2.86	4.38	3.10	6.29	2.97	5.12
BN✓ MP×	2.69	4.46	2.49	3.71	2.98	4.84	2.72	4.34
BN× MP✓	3.28	5.54	2.88	4.30	3.21	6.29	3.12	5.38
BN✓ MP✓	4.64	7.65	4.82	7.54	4.44	8.36	4.63	7.85

**Table 5 sensors-25-01070-t005:** Average localization errors of 25 human skeletal joints for models trained using different module.

Module	*x* (cm)		*y* (cm)		*z* (cm)		Average (cm)	
	MAE	RMSE	MAE	RMSE	MAE	RMSE	MAE	RMSE
×	5.83	7.70	4.78	6.21	6.20	9.07	5.60	7.66
GRU	3.09	4.98	2.74	3.99	3.30	5.32	3.04	4.76
MLP	3.90	5.49	3.28	4.39	4.06	5.70	3.75	5.19
Transformer	3.28	4.90	2.77	3.83	3.46	5.39	3.17	4.71
BiGRU	2.69	4.46	2.49	3.71	2.98	4.84	2.72	4.34

## Data Availability

The data used in this paper are collected through our own experiments and are not yet publicly available. However, data may be obtained from the authors upon reasonable request.

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
