# Peer review of "A Sliding Window-Based CNN-BiGRU Approach for Human Skeletal Pose Estimation Using mmWave Radar"

_sensors, 2025, doi:10.3390/s25041070_

Round 1
Reviewer 1 Report
Comments and Suggestions for Authors
This paper presents a CNN-BiGRU based method for human skeletal pose estimation using mmWave radar, aiming to improve accuracy by incorporating spatial (CNN) and temporal (BiGRU) modeling. The proposed approach is compared with existing methods such as MARS and mm-Pose, and the results demonstrate lower mean absolute error (MAE) in joint localization.
While the paper introduces a sequential modeling component (BiGRU) to improve motion tracking, there are concerns regarding substantial similarities with the MARS system (ACM TECS, 2021) in terms of data processing pipeline, input representation, evaluation metrics, and benchmarking methods. The main contribution of the paper appears to be the addition of BiGRU, but beyond this, the degree of novelty remains unclear.
To ensure originality and research value, the authors should explicitly differentiate their work from MARS, provide a clear discussion of their novel contributions, and enhance experimental validation by including comparisons with more recent transformer-based approaches.
Areas for Improvement and Major Concerns
1. High Similarity with MARS (ACM TECS, 2021)
The proposed CNN-BiGRU approach shares numerous similarities with MARS, including: Both papers focus on human skeletal pose estimation using mmWave radar; Both use point cloud features; Similar sorting and transformation of mmWave point cloud data; Both use Mean Absolute Error (MAE) as the primary accuracy measure.
It is acceptable to refer to existing papers when writing, but complete replication is not permitted. This is particularly important in Part 3, where FMCW MIMO mmWave radar point cloud signal processing encompasses various theories. Directly copying others' work is strictly prohibited.
The authors must explicit MARS work and provide a detailed discussion on how their method improves upon it. If this paper is an extension of MARS, it should be clearly stated, and substantial additional contributions should be demonstrated.
2. Unclear Novelty Beyond BiGRU Integration
The only major methodological change compared to MARS is the addition of BiGRU. While BiGRU enhances temporal modeling, it alone does not constitute a sufficiently novel contribution.
Recommendation:
Compare with state-of-the-art transformer-based models (e.g., STAPFormer, HRPVT, EHGFormer) to further verify the effectiveness of BiGRU.
Add occlusion and cross-environment testing to demonstrate model robustness.
3. Data Source and Reproducibility Issues
The dataset (44,781 frames, 10 motion types) is very similar to the MARS dataset (40,083 frames, 10 movements). The authors must explicitly describe the dataset collection process, and is the dataset publicly available for reproducibility?
4. Lack of Real-Time Performance Analysis
Paper does not analyze computational efficiency or latency. The authors should provide inference time and compute overhead estimates, including latency (ms/frame), FLOPs, and energy consumption analysis.
5. Need for Broader Experimental Validation
Compare the robustness of the model against external data sets, such as those provided by MARS.
Minor Suggestions
- Add a table summarizing radar system parameters – The paper currently lacks a detailed specification of the radar system used. Including a table listing key parameters (e.g., frequency, resolution, range) would improve clarity.
- Clarify the number of volunteers in the data collection section – The paper does not specify how many participants were involved in the data acquisition process. Providing this information would enhance reproducibility and help assess dataset diversity.
- Specify the exact poses analyzed in Figure 10 – The description of Figure 10 should explicitly mention which specific postures or movements are being evaluated to avoid ambiguity.
- Consider renaming "Table 3" on Page 13 to "Figure 11".
Reviewer 2 Report
Comments and Suggestions for Authors
This paper is written well and interesting.
Please find the following minor comments:
1. What is the sampling frequency of the mm-wave radar?
2. What is the calculation time? Considering the high potential use in field, time consuming is an essential factor?
3.Line 227, "the radar captures the first 64 reflection points", why 64? what does this mean?
4. To make readers understand better, please give more information in Figure 3. e.g., in the middle figure, what is the difference among those six groups? and the dash line square means what?
5. I did not see any information about participants? How many involved?
6.Can the accuracy be affected by human body type? like tall vs short, thin vs fat? How about the cloth?
Reviewer 3 Report
Comments and Suggestions for Authors
The authors propose a mmwave radar based pose tracking system and implement a new learning model called CNN-BiGRU for the pose generation. The experiments are conducted to demonstrate that the system could achieve better performance than the traditional pose tracking systems.
Strength:
The learning model is newly proposed for the pose tracking based on the mmave radar.
The new data preprocessing data method is suggested in the paper to reduce the error in 3D coordinates of skeletal points.
Weakness:
1. The mmwave radar based pose tracking system is already proposed few years ago. In CVPR 2024, the authors have proposed a WiFi based 3D pose tracking system for indoor environment, which is much more complicated than the mmave radar approach. The authors should discuss the existing works and further elaborate on the novelty of the paper.
2. The model of used in the paper is a simple combination of CNN and BIGRU. The benefits of the new proposed model should be further illustrated
3. The main contribution of the paper is the better accuracy of pose estimation. However, the authors should provide more details for the experiments to demonstrate the superiority. Firstly, the experiments environment is the Anechoic Chamber, which is too clean for the pose estimation. Besides, the skeleton joints used for different method are different. More solid results should be provided in the paper.
Round 2
Reviewer 1 Report
Comments and Suggestions for Authors
The author has addressed all my concerns well. I have no further comments and support the publication.
Reviewer 3 Report
Comments and Suggestions for Authors
The paper is good to be published.